# The evolutionary maintenance of Lévy flight foraging

**Winston Campeau**[1]*, **Andrew M. Simons**[1], **Brett Stevens**[2]

**1** Department of Biology, Carleton University, Ottawa, Ontario, Canada, **2** School of Mathematics and Statistics, Carleton University, Ottawa, Ontario, Canada

* winston.campeau@carleton.ca

## Abstract

Lévy flight is a type of random walk that characterizes the behaviour of many natural phenomena studied across a multiplicity of academic disciplines; within biology specifically, the behaviour of fish, birds, insects, mollusks, bacteria, plants, slime molds, t-cells, and human populations. The Lévy flight foraging hypothesis states that because Lévy flights can maximize an organism's search efficiency, natural selection should result in Lévy-like behaviour. Empirical and theoretical research has provided ample evidence of Lévy walks in both extinct and extant species, and its efficiency across models with a diversity of resource distributions. However, no model has addressed the maintenance of Lévy flight foraging through evolutionary processes, and existing models lack ecological breadth. We use numerical simulations, including lineage-based models of evolution with a distribution of move lengths as a variable and heritable trait, to test the Lévy flight foraging hypothesis. We include biological and ecological contexts such as population size, searching costs, lifespan, resource distribution, speed, and consider both energy accumulated at the end of a lifespan and averaged over a lifespan. We demonstrate that selection often results in Lévy-like behaviour, although conditional; smaller populations, longer searches, and low searching costs increase the fitness of Lévy-like behaviour relative to Brownian behaviour. Interestingly, our results also evidence a bet-hedging strategy; Lévy-like behaviour reduces fitness variance, thus maximizing geometric mean fitness over multiple generations.

## Author summary

In heterotrophs, incuding animals, survival depends on the net energy gained through foraging. The expectation, then, is that natural selection results in adaptations for efficient foraging that optimize the balance of searching costs and rewards. Lévy flight foraging has been proposed as an optimal foraging solution. The hypothesis states, if no information about resource locations are available, and the locations are re-visitable, then selection will result in adaptations for Lévy flight foraging, a type of random walk. It has been argued that Levy-like foraging behaviour may simply reflect how resources are distributed, but empirical and theoretical research suggests that this behaviour is intrinsic or innate. However, this research does not address evolutionary mechanisms, and lacks ecological

**Data Availability Statement:** All relevant data are within the manuscript and its Supporting information files.

**Funding:** This work was supported by the Natural Sciences and Engineering Research Council of Canada: NSERC DG 2021-03832 to AMS, and the

Natural Sciences and Engineering Research Council of Canada: NSERC RGPIN 06392 to BS. The funders had no role in study design, data collection and analysis, decision to publish, or preparation of the manuscript.

**Competing interests:** The authors have declared that no competing interests exist.

breadth. We extend the current theoretical framework by including evolutionary ecological contexts. We treat an organism's random walk as a heritable trait, and explore ecological contexts such as population size, lifespan, carrying capacity, searching costs, reproductive strategies, and different distributions of food. Our evolutionary simulations overwhelmingly resulted in selection for Lévy-like foraging, regardless of the distribution of food, and evidences Lévy flight foraging as a bet-hedging strategy. Thus, here we provide some of the first evidence for the evolutionary maintenance of Lévy flight foraging.

## Introduction

A Lévy flight can be described as a random walk with move lengths pulled from a heavy-tailed distribution, $P(l) \sim l^{-u}$ with power-law exponent $1 < u < 3$ [1–4] (Fig 1). Levy flights model the behaviour of phenomena in several areas including, but not limited to, finance [5–7], criminology [8, 9], optimization algorithms and cryptography [10–13], epidemiology [14–16], earthquake analysis [17], and physics and astronomy [18–22]. In behavioural ecology, Levy flights have also been shown to be an efficient searching strategy for foragers [2–4]. Efficiency of the search for food is given by the balance between cost and reward, and is key to survival [23–26]. If the distribution of resources is unknown, the resources are sparse, randomly distributed and revisitable, and the searcher has no memory, then the Lévy flight foraging hypothesis states that a Lévy flight with power-law exponent $u \simeq 2$ is an optimal or near-optimal searching strategy. There is ample evidence demonstrating the prevalence of 'Lévy-like'

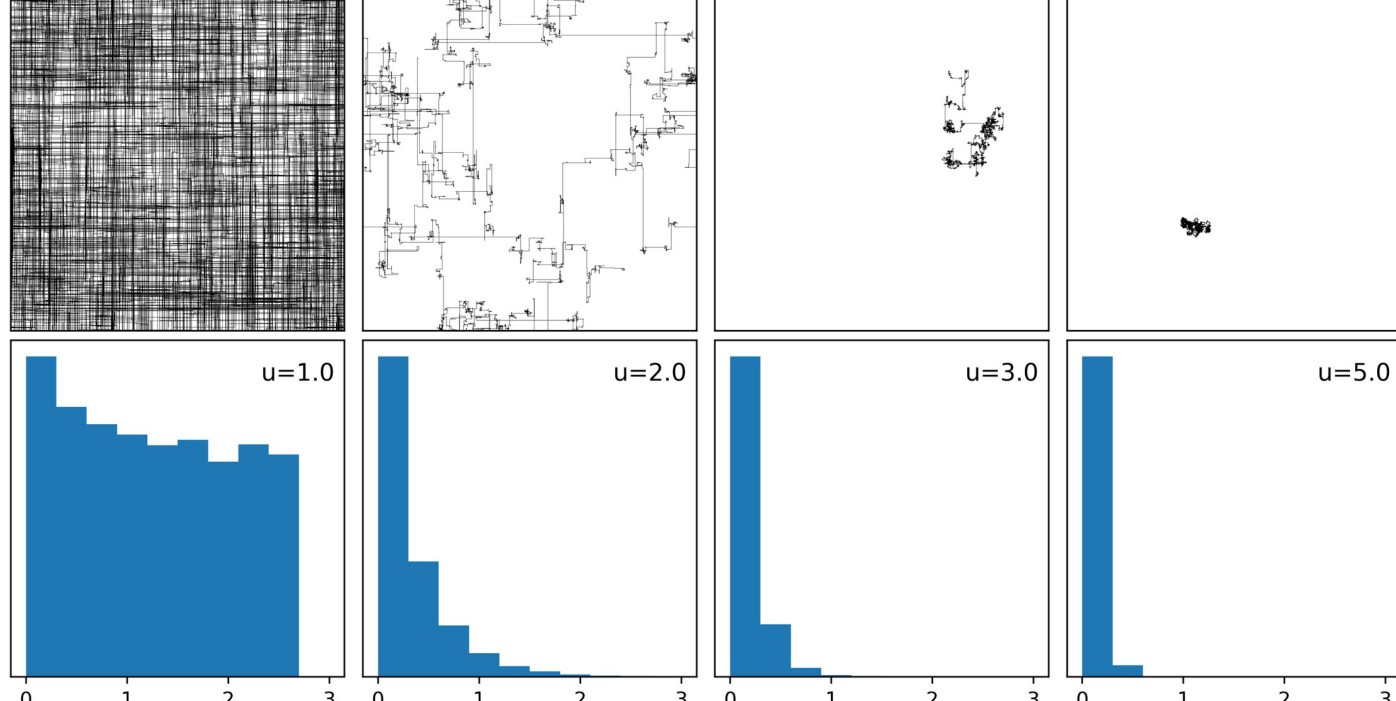

**Fig 1. Examples of random walks.** Four random walks (top) of $10^4$ segments with move lengths pulled from a discrete truncated power-law distribution with exponents $u$ = 1.0, 2.0, 3.0 and 5.0, with random direction from $\{0, \pi/2, \pi, 3\pi/2\}$. The walks begin at a random location on a $1000 \times 1000$ toroidal environment and their power-law distribution (bottom) is shown as a histogram of $10^4$ logged move lengths truncated to half the length of the environment. Walks with exponents near $u = 1.0$ exhibit ballistic movement, diffusive movement near $u = 2.0$, and superdiffusion for $u \geq 3$.

behaviour in the foraging and movement patterns among biological units, including species of fish, birds, insects, mollusks, slime molds, t-cells, bacteria, plants, human populations, and in the fossil records of 50 My-old sea urchins [27–38]. Although prevalent, whether Lévy-like behaviour is the result of selection for behavioural adaptations (the intrinsic or adaptationist hypothesis) or is an emergent phenomenon due to the encounters within an environment's distribution of resources (the extrinsic or emergentist hypothesis) remains a topic of debate [30, 39]. An additional source of debate is whether Lévy strategies are always advantageous over Brownian strategies [2, 4, 40, 41]. If Lévy flight is an optimal or near-optimal searching strategy, and a heritable trait, then adaptations for Lévy flight foraging should result from natural selection.

Perhaps the most compelling empirical evidence for the intrinsic hypothesis is the demonstration of Lévy-like behaviour in 'brain-blocked' Drosophila larvae [30]. The mean power-law exponent of larvae with blocked synaptic activity, suboesophageal ganglion, and sensory neurons was found to be 1.96, suggesting a selective pressure for autonomous Lévy walks close to the theoretical optimum. And on the theoretical end, using reaction-diffusion algorithms, Lévy walks have been shown to be an optimal solution across a broad range of environments [4]. Since the extrinsic hypothesis relies on searching strategies specific to distinct environments, the broad success of Lévy walks counters this argument. Another theoretical approach incorporates spatial memory (via Gaussian mixture models) with a diffusion model, where the strength of memory effects resulted in the proclivity to not leave the first few visited sites, abandoning Lévy-like behaviour and favouring Brownian behaviour [42]. If resource locations are static and revisitable, then strong spatial memory is a sufficient strategy as an organism can revisit known resources indefinitely. However, in a fluctuating environment strong spatial memory may lose its utility if the distribution of resources is continually changing. Given that a temporally fluctuating environment is inextricably linked with evolution [43–48], there are arguably cases where more explorative searching behaviour, due to weaker memory, would lead to higher fitness outcomes over evolutionary time. And if strength of memory is a heritable trait, the selective pressure for a weaker spatial memory—or Lévy-like behavior—would be higher, thus providing additional argumentation for the intrinsic hypothesis.

These theoretical models have advanced our understanding of the advantages of Lévy-like behaviour, but provide an incomplete and only proximate explanation. In order to support the claim of an evolutionary origin we must first address several key aspects of evolutionary ecology. These aspects can be divided into two main categories, evolutionary mechanisms and ecological contexts. Much of the theoretical research on Lévy walks concerns itself more with physical explanations [4, 40, 42] and does not include evolutionary mechanisms. The intrinsic hypothesis relies on the selection and heritability of an organism's distribution of move lengths; thus, testing the hypothesis should consider at least these two factors. A more complete understanding of the evolution of a foraging behaviour depends on the success under ecological contexts such as energetic costs and reproductive strategies. To determine whether selection will result in adaptations for Lévy-like behaviour, we test the Lévy flight foraging hypothesis using numerical simulations that include biological contexts such as population size, searching costs, lifespans, resource distribution, speed of movement, competition, and different measures of energy accumulation as proxies for semelparity and iteroparity. Our evolutionary models include clear aspects of selection & heritability, and utilize the searching costs and rewards of foraging as the medium through which selection acts on foraging behaviour. Thus, if Lévy-like foraging behaviour is both adaptive and instrinsic, then our simulations should result in the selection and evolution of Lévy-like behaviour, over a broad array of conditions. Whereas if Lévy-like foraging behaviour is extrinsic, the expectation is that selection

will result in foraging exponents that are positively correlated with the spatial exponent governing the distribution of resources.

## Materials and methods

We used a combination of two methods of simulation to test the Lévy flight foraging hypothesis; a set of selection simulations which probe the parameter space to determine fitness curves over a single generation, and lineage-based models of evolution which test subsets of parameters determined by the single-generation simulations to be of interest. Both methods utilize matrix environments with resources distributed according to Lévy dust (LD) and uniform random patterns. Digital organisms (DO) with variable power-law exponents of move lengths traverse these environments until they have exhausted their lifespan, during which their energy yield, our surrogate for fitness, is computed. We then determine the optimal exponent of a population of DOs by comparing calculations of their resultant energies.

### Generation of environments

The model environment is a $n \times n$ toroidal matrix (Fig 2) (i.e. if a DO travels over an edge of the matrix, then it would appear on the opposite edge), $E$, with resource entries $e_{i,j}$ $i, j \in \mathbb{Z}_n$ with $n = 1000$, which are all initialized with zeros. Two methods were used to populate the matrix with resources; uniform random distributions and Lévy dust, a fractal point pattern. If a location $(i, j)$ is selected during resource distribution, then $e_{i,j}$ is incremented by 1. Each environment recieves exactly $n^2 \cdot 10^{-4}$ re-visitable resources; this number corresponds to the

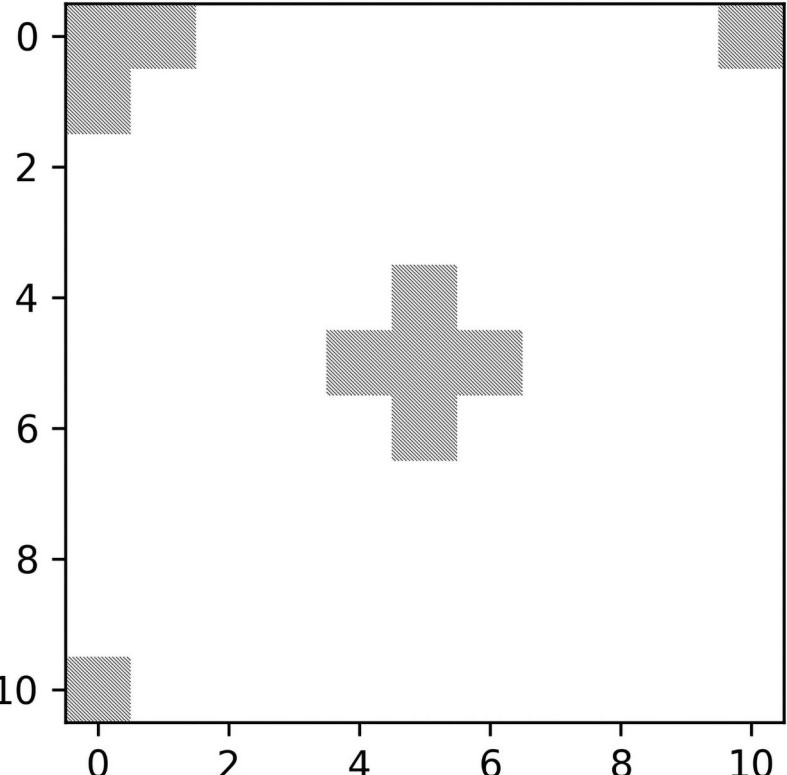

**Fig 2. Demonstration of a toroidal environment.** A set of five non-empty resource entries in a cross formation is centered at entry $e_{0,0}$ and then translated to $e_{5,5}$ on a $11 \times 11$ toroidal matrix environment.

density of resources used by Wosniak et al. [4], where exponent $u \sim 2$ was found to maximize search efficiency, regardless of resource distribution.

**Uniform random distribution of resources.** We include environments with uniform randomly distributed resources in our simulations because this was the focal type of environment studied in the seminal work for the Lévy flight foraging hypothesis [3]. Uniform distributions also describe several species' distributions; some examples are trees, shrubs, marine invertebrates, and three-spined sticklebacks [49–53]. Patches are distributed by taking a random uniform sample, with replacement, of all possible locations on the matrix. The resource entry at each selected location is then incremented (Fig 3). There is also the choice of random uniform sampling without replacement, but one-hundred locations sampled out of one-million is unlikely to differ significantly in composition from that of samples without replacement, and sampling with replacement is also consistent with the Lévy dust distributions which can result in $e_{i,j} > 1$.

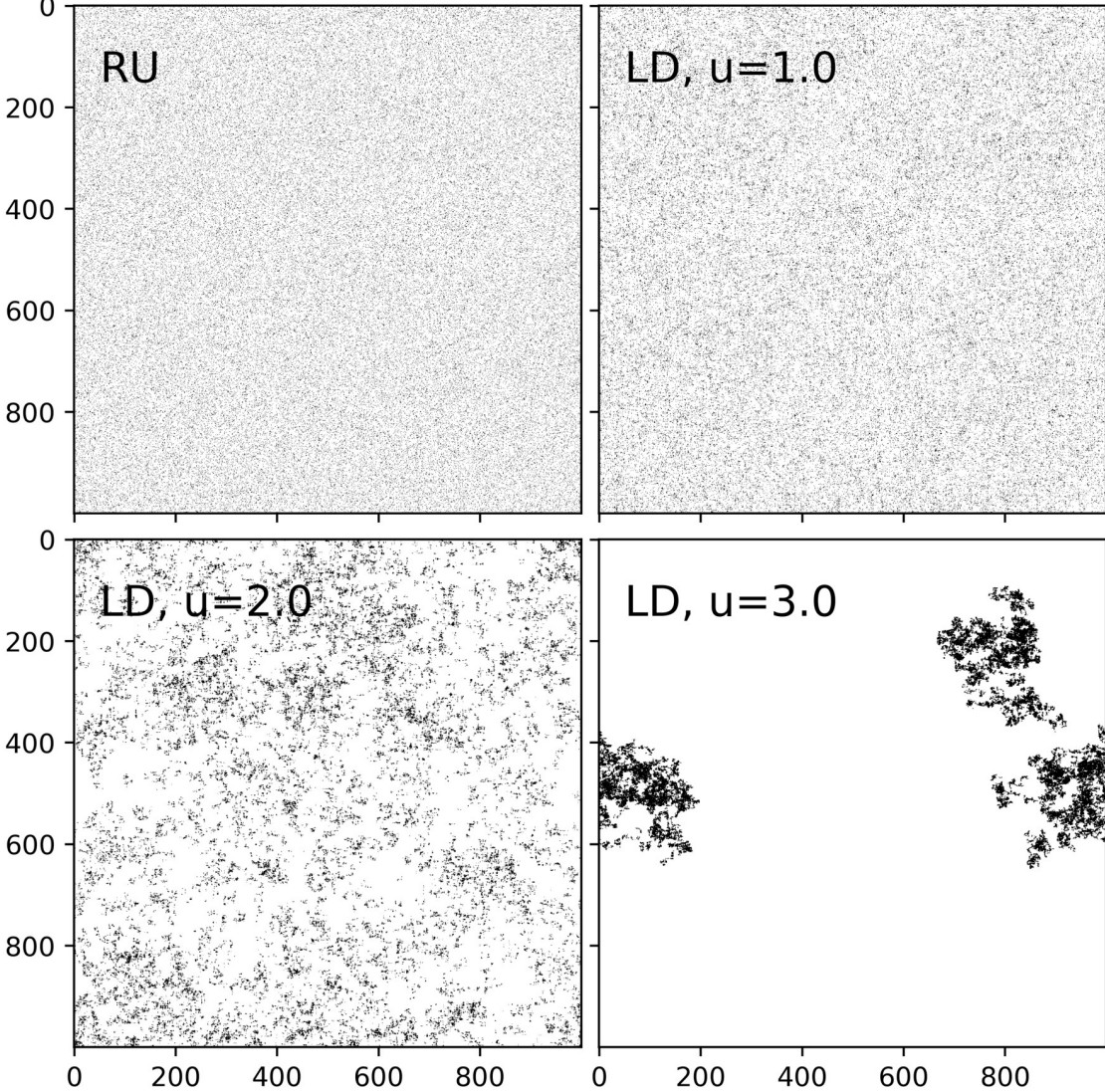

**Fig 3. Uniform random and Lévy dust environments.** Examples of $1000 \times 1000$ environments with $10^5$ uniform random, Lévy dust dimensions $u = 1.0$, $2.0$, and $3.0$ distributed resources. There are $10^5$ patches instead of $n^2 \cdot 10^{-4}$ to more clearly reveal the nature of the distributions. Note that the homogeneity of patch distribution decreases with increasing $u$.

**Lévy dust.** Lévy dust (LD) is a distribution which generates a fractal pattern known to mimic resource dispersal in nature, including at least plants [54, 55] and marine organisms [27, 56] and was utilized in perhaps the most comprehensive evolutionary study of Lévy walks [4]. For a LD environment, patches are placed using a Lévy flight with successive move lengths, l, selected from a distribution defined by the following probability mass function (pmf):

$$P(l) = \begin{cases} 0 & l < 1 \ or \ l > n/2 \\ l^{-u} & 1 \le l \le n/2 \end{cases} \tag{1}$$

where n is the size of the environment, and $u - 1$ is the fractal dimension (Fig 3). In previous studies, each move length is selected with a uniform random direction $\theta$ from $[0, \pi)$, but our model is discrete and the DOs move perpendicular to the axes; thus, we sample a random direction from $\{0, \pi/2, \pi, 3\pi/2\}$ so that both the DO walks and the environments they traverse share the same generative mechanism. However, the distribution of directions will still approach a uniform distribution over an increasing number of consecutive moves (Fig 4). We chose the upper limit of $l = n/2$ in Eq (1) as that is the maximum distance for movement perpendicular to the axes in an $n \times n$ toroidal environment.

## Simulation methods

We simulate populations of DOs. An individual DO is defined by the following list:

$$[S, \xi, \alpha\xi, i, j, \lambda, d, \alpha d]$$

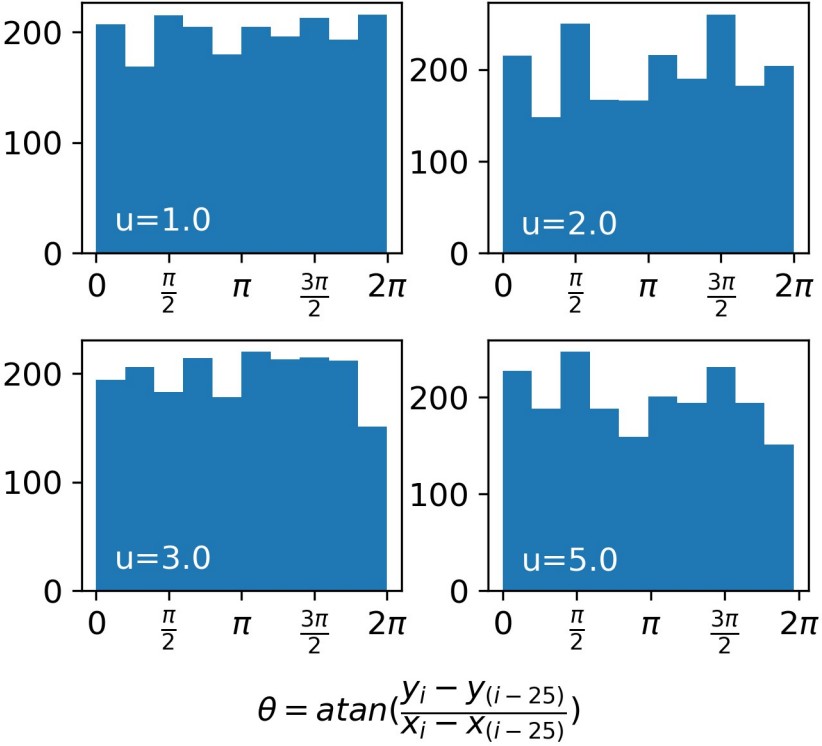

$$\theta = atan(\frac{y_i - y_{(i-25)}}{x_i - x_{(i-25)}})$$

**Fig 4. Continuity of $\theta$ over consecutive moves.** Distributions of direction, $\theta$, for random walks with power-law exponents $u$ = 1.0, 2.0, 3.0, and 5.0, within windows of 25 consecutive moves, over $10^4$ moves.

$S$ is a list of $10^4$ integer move lengths generated by the pmf described in Eq 1, with exponent, $u$, and $n = 1000$. The current amount of resources a DO has encountered, $\xi$, and the sum of resource encounters since birth, $\alpha\xi$, are used to compute the metrics of search efficiency and fitness in our evolutionary models. A DO's energy starts at zero, increases by $e_{i,j}$ upon finding a resource at position $(i, j)$, and decreases with movement by a fixed searching cost $\chi$. The parameters $i$ and $j$ are a DO's position in the environment. All DOs use a truncated random walk with movement perpendicular to the axes to search their environment; they draw a random move length from $S$ and a random direction from $\{0, \pi/2, \pi, 3\pi/2\}$, and continue moving along the environment, entry by entry, until they have either exhausted their move length or have located a resource. A search starts at a random location in the environment $(i, j)$ which is updated after every move. The lifespan, $\lambda$, is the amount of time a DO can travel before it is removed from a simulation. The total distance a DO has traveled over its lifespan to date is denoted by $d$, and the sum of those distances since birth by $\alpha d$. The variables $\xi$, $\alpha\xi$, $d$, $i$, $j$ and $\alpha d$ are functions of time and will be subscripted by $t$ when specifying the time is necessary.

We vary the searching cost to simulate a range of conditions from those where cost is extremely low (e.g. wandering albatross [57, 58]), to those where the energetic requirements of movement may be higher (e.g. insect flight or movement in denser mediums, such as water). We compute a DO's fitness in two ways, 1) using energy at the end of a lifespan (EOL); a proxy for fitness assuming semelparity or a single reproductive event just before death, and 2) using the average of energy over a lifespan (AOL); a proxy for lifetime reproductive success assuming iteroparity, or multiple reproductive events over a lifetime. To illustrate the differences between the two fitness metrics, and demonstrate how they correspond to semelparity and iteroparity, we provide the following example. If two DOs traveled the same total distance over their lifespan and encountered 100 patches each, then their fitness as determined by the EOL metric would be equivalent. However, one of the DOs might have found those 100 patches at the very end of its lifespan, while the other encountered its 100 patches early in its lifespan. The fitness of a DO with the earlier encounters would be higher by the AOL metric, as it has had access to the same resources for longer, thus with respect to iteroparity, the possibility of more reproductive events over a lifespan.

Searching cost is applied in a post-processing round after each generation. We include a short proof (Box 1), and a detailed proof (S1 Appendix) that this is equivalent to running simulations where a cost is applied at each timestep; thus, computational cost can be reduced. We denote a DO's speed as the distance traveled per timestep by $s$, the number of resources encountered at timestep $t$, by $e_t$ (a resource entry with temporal notation now, rather than spatial), the distance traveled in a timestep by $\delta_t$, and set the rule that a move of length $l$ takes $\lceil l/s \rceil$ timesteps to traverse.

**Single-generation simulations.** The single-generation simulations consisted of $5 \cdot 10^4$ DOs with power-law exponents drawn from a continuous uniform distribution on $[0, 6]$. Choosing from this range permitted exploration of the fitness outcomes of both Lévy-like exponents, $1 < u < 3$, and Brownian-like exponents, $u \geq 3$. These simulations were restricted to a single generation with fixed patch distribution types, speed, and lifespan. The following steps describe how the simulation then proceeds:

1. E is constructed with n = 1000 and populated with $n^2 \cdot 10^{-4}$ re-visitable patches.

2. Each of the $5 \cdot 10^4$ DOs traverse a unique copy of E until their lifespan is exhausted.

3. The DOs' $u$, $\xi$, $\alpha\xi$, $d$, and $\alpha d$ are saved for post-processing.

The results of these simulations are analyzed in two ways. Cost is applied post-hoc for both analyses, and any DO with $\epsilon \leq 0$ is considered dead, thus is not admitted. The first method of

### Box 1. Searching costs are a linear transformation of zero cost models

#### Proposition

Both of a DO's EOL and AOL energy can be computed from $s$, $\lambda$, $\xi_\lambda$, $\alpha\xi_\lambda$, $d_\lambda$, $\alpha d_\lambda$, for any cost $\chi$.

#### Proof

The energy of a DO at time $t$ is $\epsilon_t$, and for $t = 0$, $\epsilon_0 = 0$. When speed $s = 1$, moving a distance of one consumes exactly one timestep. Thus, if $d_{t+1} = d_t + \delta_t$ and $s = 1$, then $\delta_t = 1$ and $d_t = t \; \forall t$. The DO's energy is updated by,

$$\epsilon_{t+1} = \epsilon_t + e_t - \chi\delta_t = \epsilon_t + e_t - \chi$$

By letting $\xi_t = \sum_{\tau=0}^{t} e_\tau$ be the sum of resource entries found at time $t$ since birth, the energy at the end of a DO's lifespan, $\epsilon_{EOL}$, can be described as:

$$\epsilon_{EOL} = \sum_{t=0}^{\lambda}(e_t - \chi) = \xi_\lambda - \lambda\chi$$

and $\epsilon_{AOL}$ as:

$$\epsilon_{AOL} = \frac{1}{\lambda}\sum_{t=0}^{\lambda}\epsilon_t = \frac{1}{\lambda}\sum_{t=0}^{\lambda}\sum_{\tau=0}^{t}(e_\tau - \chi) = \frac{1}{\lambda}\alpha\xi_\lambda - \frac{\chi}{\lambda}\frac{\lambda(\lambda+1)}{2}$$

Speeds of $s > 1$ result in $\delta_t > 1$, where generalization for any $s$ is achieved by tracking the distance a DO has traveled:

$$\epsilon_{EOL} = \epsilon_\lambda = \sum_{t=0}^{\lambda}(e_t - \chi\delta_t) = \xi_\lambda - d_\lambda\chi$$

$$\epsilon_{AOL} = \frac{1}{\lambda}\sum_{t=0}^{\lambda}\epsilon_t = \frac{1}{\lambda}\sum_{t=0}^{\lambda}\sum_{\tau=0}^{t}(e_\tau - \chi\delta_\tau) = \frac{1}{\lambda}\alpha\xi_\lambda - \frac{\chi}{\lambda}\alpha d_\lambda$$

Thus, models with $\chi \neq 0$ remain a simple linear transformations of $\chi = 0$ models, with the additional requirement of recording $d_\lambda$ and $\alpha d_\lambda$ for speeds of $s > 1$.

See supplementary information (S1 Appendix) for a more detailed proof.

analysis is plotting the entirety of the DO population's fitness as a function of their exponents, along with a sliding window of the energy's first moment, positive-only first moment, and standard deviation. We use a positive-only first moment to ensure only DO's with fitness $\epsilon > 0$ are considered. The second is sampling 1000 sub-populations each of sizes 10 through 1500 with step-sizes of 10 and extracting the exponents of the top 1% performing DOs from each sample. These two analyses were used to provide insight on which subsets of the parameter space should be tested with evolutionary simulations.

**Evolutionary simulations.** The evolutionary simulations are lineage-based and multi-generational extensions of the single-generation simulations, where the power-law exponent of a DO's $S$ is treated as a heritable and normally distributed trait. A DO's fitness is measured either by $\epsilon_{EOL}$ or $\epsilon_{AOL}$, and the composition of subsequent populations was determined by relative fitness. The following is a step-by-step outline of how the evolutionary model operates:

1. Initialization

   1.1. Select searching cost $\chi$, number of generations $G$, resource distribution type, population size $K$ (also the carrying capacity), and speed $s$.

   1.2. Generate the starting population of size $K$ with exponents $u \in N(\mu, \sigma_{sv})$ where $\mu$ is the mean foraging exponent of the population prior to evolution and $\sigma_{sv}$ is the standing variation.

2. DO Search

   2.1. In each generation, an $E$ is constructed with $n = 1000$ and populated with $n^2 \cdot 10^{-4}$ revisitable patches.

   2.2. The population of DOs traverse E until all lifespans are exhausted.

   2.3. Searching costs are then applied to each DO.

3. Relative Fitness Function

   3.1. For each DO, $i$, a triple $(u_i, f_i, \omega_i)$ is stored, where $u$ is the power-law exponent, $f$ is the fitness, and $\omega$ is the number of offspring. Initially, $\omega_i = 0$.

   3.2. Any triple $(u_i, f_i, \omega_i)$ with $f_i \leq 0$ is removed. If all $f_i \leq 0$, the simulation ends. Otherwise,

   3.3. the remaining triples are stored in descending order of $f_i$. This is the parent list, $\Omega$ of length $m$.

   3.4. The number of offspring are then assigned by $\omega_i = \lceil \frac{K \cdot f_i}{\sum_{j=0}^{m} f_i} \rceil$, until $\sum_{i \in \Omega}^{m} \omega_i = K$.

4. Next Population

   4.1. For each parent DO, $0 \leq i \leq m$, $\omega_i$ offspring DOs are created with $u = u_i + N(0, \sigma_{sv})$

   4.2. The previous generation is replaced by its offspring.

5. Repeat steps 2)-4) $G$ times.

To elucidate steps 3) & 4), we give a brief example. If a simulation after steps 1) and 2) with $K = 4$ resulted in the following set of DOs being passed to the relative fitness function {$(u = 3.4, f = 7), (u = 5.0, f = 2), (u = 2.3, f = 2), (u = 1.5, f = -1)$}, then the respective number of offspring assigned to each DO would be 3, 1, 0, and 0. Each offspring's foraging exponent is their parent's exponent, plus a random normal value with mean zero and $\sigma_{sv}$. The extreme case is $\sigma_{sv} = 0$ in which offspring are exact copies of their parents; thus all pheontypic variance is due to genetic variance among DO lineages and heritability ($h^2$) = 1. With increasing $\sigma_{sv}$, $h^2$ is effectively reduced, and offspring are increasingly likely to look dissimilar to their parents. The subsets of the parameter space extracted from the single-generation simulations were combined with $\sigma_{sv}$ to explore the differential survival and evolution of DO's due to differences in their power-law exponents.

Relative performance under direct competition is an important component of evolution; thus, competitive assays provide additional insight into fitness [59]. Importantly, they can help

discern whether an evolved trait has any biologically significant fitness advantages over an ancestral trait. Thus, we included a set of simulations in which populations of fixed foraging exponents competed for a fraction of a carrying capacity. The competition simulations operate similarly to the evolutionary simulations, but with two main differences, 1) they start with two equally-sized populations, each with a unique and fixed foraging exponent, and 2) $\sigma_{sv} = 0$.

**Determining the relationship between $\epsilon_{EOL}$ & $\epsilon_{AOL}$.** We wanted to determine the relationship between our two measures of fitness, $\epsilon_{EOL}$ and $\epsilon_{AOL}$, the average energy over a lifespan and the energy at the end of a lifespan, respectively. Given a strong relationship, some computational time could be saved by focusing on a single measure of fitness. We compared the results of $5 \cdot 10^4$ DO's $\epsilon_{AOL}$ and $\epsilon_{EOL}$ for lifespans $\lambda$ = 1M, 2.5M, 5M, 10M, for $s$ = 1 and $\chi$ = 0, across all environments with $n^2 \cdot 10^{-4}$ re-visitable patches using simple linear regression. The Thiel-Sen estimator, a robust form of simple regression, was also used to evaluate comparisons as some of the results had visible skew and heteroskedasticity.

Among the regression models, all coefficients of determination fell within 0.76 and 0.87 with $p << 0.05$. The coefficients of determination consistently increased with lifespan, which is likely due to the probabilistic nature of the searching behaviours; as searching time increases, sampling time increases, thus the error in the estimated slope decreases. Regardless of environment, lifespan, or statistical estimator, a slope of $\sim 0.5$ was consistently reported. A slope of 0.5 likely implies that resource discovery is balanced over a lifespan. As an example, say a DO travels a distance $n$ over its lifespan, and encounters a resource entry every $k$ steps. Assuming $k$ divides $n$, then the DO will visit exactly $n/k$ resource entries and,

$$\epsilon_{EOL} = n/k$$

$$\epsilon_{AOL} = \frac{\sum_{i \in n} \epsilon_{EOL_i}}{n} = \frac{n+2}{2k} - \frac{1}{2}$$

When $n >> k$, the ratio $\epsilon_{AOL}$: $\epsilon_{EOL}$ will approach $\sim 0.5$, and will linearly depart from 0.5 as $k$ approaches $n$. As an example, consider $k = 10^4$ and the shortest lifespan examined $\lambda$ = 1M, then the ratio of the two measures is $\sim 0.495$. This result is commensurate with the average result of Brownian-like DOs with lifespans of $\lambda$ = 1M with an average $\epsilon_{EOL} \simeq 100$, and average $\epsilon_{AOL} \simeq 50$, thus a ratio of $\sim 0.5$ (S1 Fig). For contrast, if a DO found all $n/k$ resources in sequence at the start or end of its lifespan, then

$$\epsilon_{AOL} = \begin{cases} start, & \dfrac{n - \dfrac{n}{k}}{k} + \dfrac{\dfrac{n}{k} + 1}{2k} \\[4ex] end, & \dfrac{\dfrac{n}{k} + 1}{2k} \end{cases}$$

where in either case, the only value of $k$ which satisfies a ratio of 0.5 is $k = 1$, and values of $k > 1$ asymptotically approach 1 and 0 for the start and end cases, respectively. Due to the strong relationship between our two fitness metrics, and because reproduction strategies tend towards iteroparity [60, 61], we chose to focus on the $\epsilon_{AOL}$ fitness metric for both visualizations and evolutionary simulations. However, any notable differences in results between the two fitness measures will be pointed out when necessary.

All simulations were programmed in Python 3.7.9 using the numpy [62], scipy [63], and pandas [64] libraries in addition to the Python standard library. Parallelization was accomplished using a combination of GNU parallel [65], multiprocessing [66], and bash scripting. Data visualizations were accomplished with a combination of Matplotlib [67] and R [68]. Computations were performed on the Niagara, Graham, and Cedar supercomputers at the

HPC consortium which provided access to as many as 200 CPU cores and 875GB of RAM [69, 70]. Code for the single-generation and evolutionary simulations (S1 Code), and the results (S1 Data), are available in the supplementary information.

## Results & discussion

Lévy flights have been evidenced as an optimal, or near-optimal, method of searching for resources given that the information on the distribution of resources is unknown, the resources are sparse, randomly distributed and revisitable, and the searcher has no memory. Building on this premise, the Lévy flight foraging hypothesis states that natural selection should result in Lévy flight foraging. However, the maintenance of Lévy flight foraging through evolutionary processes is lacking in existing models, and they lack ecological breadth. Building upon existing theoretical models, we treated a distribution of move lengths as a variable and heritable trait, and added ecological contexts such as population size, searching costs, speed of movement, lifespans, reproductive strategies, and different resource distributions. The results presented here support the Lévy flight foraging hypothesis, suggest that Lévy flight foraging may be selected for over the longer term as a bet-hedging strategy, but also indicate it may not be universally optimal.

### Single-generation

Our single-generation simulations explored how fitness varied with various population sizes, foraging exponents, searching costs, speeds, and lifespans over a single generation. These simulations confirmed that Lévy-like exponents have higher mean fitness outcomes under a broad array of circumstances. A foraging exponent of $u \simeq 2$ had the highest mean fitness when there were no associated costs for searching (Fig 5), and this remained true regardless of the resource distribution, consistent with the previous findings of Wosniak [4]. However, while the simulations without searching costs consistently demonstrated Levy-like exponents had the highest mean fitness, the fitness variance of Brownian-like exponents was large enough to produce individual DOs with fitness which exceeded that of any individual Levy-like DO (Fig 6).

Once a sufficiently high searching cost is applied, the variance in fitness results in the survival primarily of DOs with high fitness Brownian exponents compared to the relatively lower fitness (although, optimal on average) Lévy-like exponents. This result is in agreement with Palyulin's 1-dimensional searching model [40], where including the effect of external drift indicated Brownian strategies as the most advantageous. However, the magnitude of this effect, here, is conditional on population size, length of lifespan, the resource distribution, and searching costs. Results from all $5 \cdot 10^4$ DOs demonstrated that the foraging exponent with the highest mean fitness tends to be Brownian for clumpy resource distributions (LD fractal dimension u = 2.0, 3.0), shorter lifespans, and high searching costs (Fig 7). Nevertheless, selection tends towards Lévy-like exponents with sufficiently long lifespans, regardless of a searching cost, and the strength of selection increases when resources are uniformly distributed. Interestingly, while both fitness metrics mainly share the same fitness outcomes, the end of lifespan fitness metric more often resulted in selection for Lévy-like exponents for lifespans $\lambda$ = 10M.

The differences in selection for Lévy-like and Brownian exponents are highlighted when subsampling from populations of $5 \cdot 10^4$ DOs, and collecting the top 1% performing DOs. Those DOs which traversed random uniform environments see increasing selection for Lévy-like exponents when lifespans are longer, population size is smaller, or the searching costs are low (Fig 8). This trend holds true for all resource distributions, but selection for Lévy-like

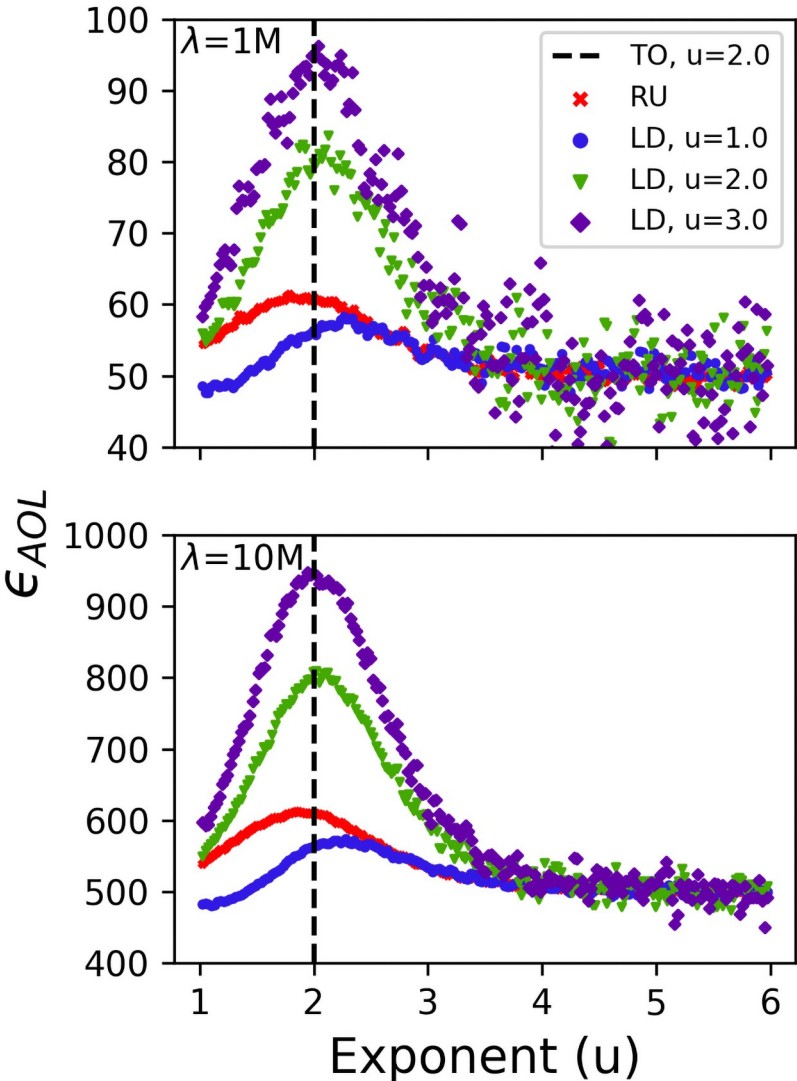

**Fig 5. DO fitness curves.** The average energy over a lifespan $\epsilon_{AOL}$, of populations of $5 \cdot 10^4$ DOs with foraging exponents $1 \leq u \leq 6$ and lifespans $\lambda = 1M$ (top) & $\lambda = 10M$ (bottom) traversing all environments types (RU = Random Uniform [red 'x's], LD = Lévy Dust u = 1 [blue circles], u = 2 [green triangles], and u = 3 [purple diamonds]) with no searching costs. The fitness trends were captured with a sliding window of first moments. The theoretical optimal foraging exponent (TO) is indicated with the vertical dashed line $u = 2.0$.

exponents is reduced for LD environments, compared to random uniform environments, and with decreasing clumpiness of resources (Fig 9).

The results presented so far can be attributed to two kinds of sampling effects: population size and lifespan. The fitness variance of Brownian-like exponents often produces more individual DOs with fitness lower than Lévy-like exponents than it does with higher fitness; thus smaller population sizes are more likely to sample low fitness Brownian DOs, which explains the fitness advantage of Lévy-like exponents for small population sizes. And the longer a DO has to traverse an environment, the more it can 'sample' the environment; thus fitness variance decreases with increasing lifespan (Figs 5 and 6), and the resultant fitness is closer to the mean fitness associated with its foraging exponent.

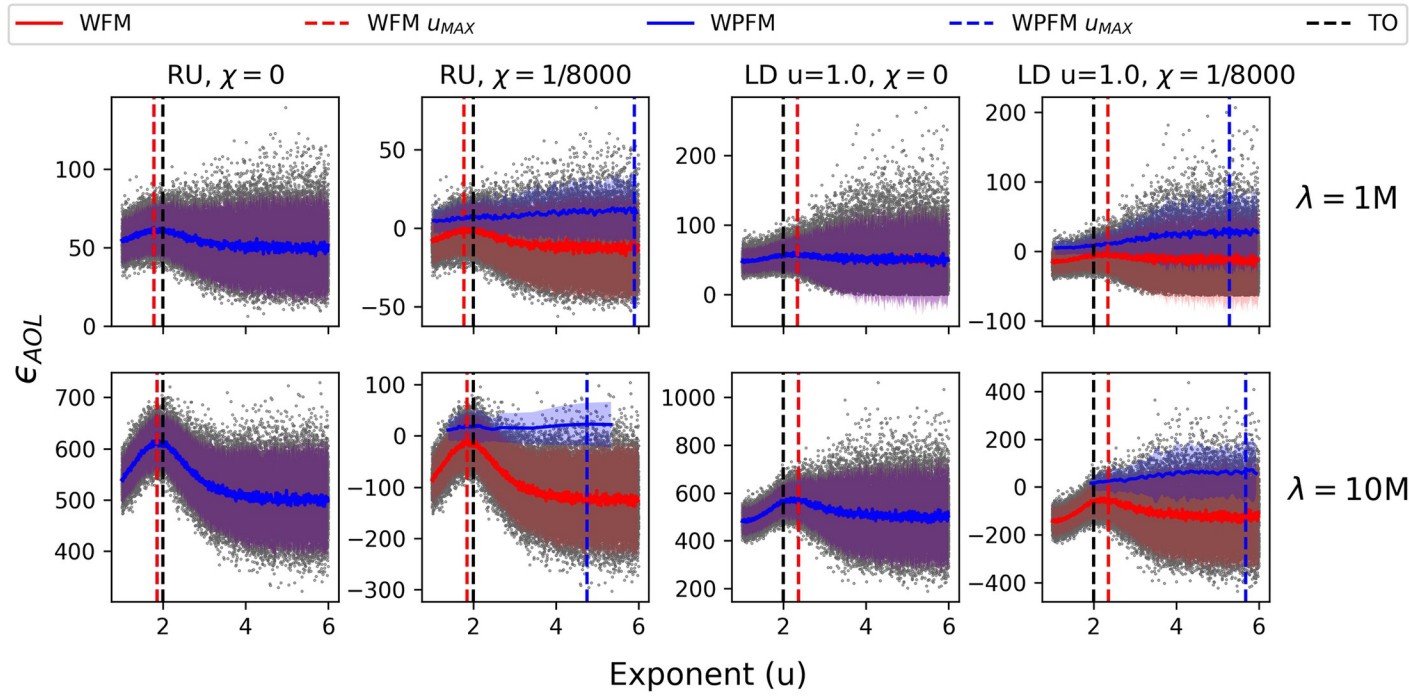

**Fig 6. Fitness variance and the effect of a searching cost.** The average energy over a lifespan $\epsilon_{AOL}$, of populations of $5 \cdot 10^4$ with foraging exponents $1 \leq u \leq 6$ and lifespans $\lambda$ = 1M & $\lambda$ = 10M traversing random uniform (RU) and Lévy dust (LD) dimension $u$ = 1.0 environments. The fitness trends were captured with a sliding window of the first moment (WFM, red) and positive-only first moment (PWFM, blue), with two standard deviations surrounding the trend. The exponent which maximizes fitness, $u_{MAX}$, is extracted from the sliding windows and marked with a dashed vertical line. Note that the first moments and standard deviations are superimposed in the zero-cost ($\chi$ = 0) plots.

A subset of our single-generation simulations explored the idea of how increasing speed might alter fitness outcomes. When speed is one, traveling a distance of length $l$ is equivalent to $l$ timesteps of a DO's lifespan, and when $s > 1$, it takes $\lceil l/s \rceil$ timesteps to travel. For simulations without cost, increasing speed results in selection for DOs with ballistic-like foraging (Fig 10). The reason ballistic DOs are advantageous in this scenario is because they are allowed to travel much further in their lifespans than Lévy-like or Brownian-like DOs. For contrast, higher foraging exponents, $u > 3$, will increasingly result in a move length distribution of all ones, while an exponent of $u$ = 1 constitutes a distribution with many move lengths larger than one (Fig 1). As speed increases, DOs with higher foraging exponents will travel a total distance approximately equal to their lifespan, whereas ballistic exponents will travel distances several-fold their lifespan, thus encountering more resources by virtue of total distance traveled alone. Adding a cost penalizes DOs differentially due to the total distance traveled corresponding to their foraging exponent. The result is peak fitness shifting towards Lévy-like exponents, and then to Brownian-like exponents, with increasing searching cost. This result assumes no metabolic or physical constraints due to increasing speed, which might be unrealistic as drag, for example, increases with speed. Our model of increasing speed can also be thought of as differences in timings in a DO's decision-making; at faster speeds, longer and shorter move lengths are traveled over the same amount of time. Brownian-like DOs would 'decide' to travel primarily distances of one, when in the same amount of time, those DOs could just as well have traveled a longer distance in increments of one. Our simulations of faster speeds deviate from the behaviour of current models, and were not included in the evolutionary simulations; however, they serve as a conceptual framework previously unexplored in the literature.

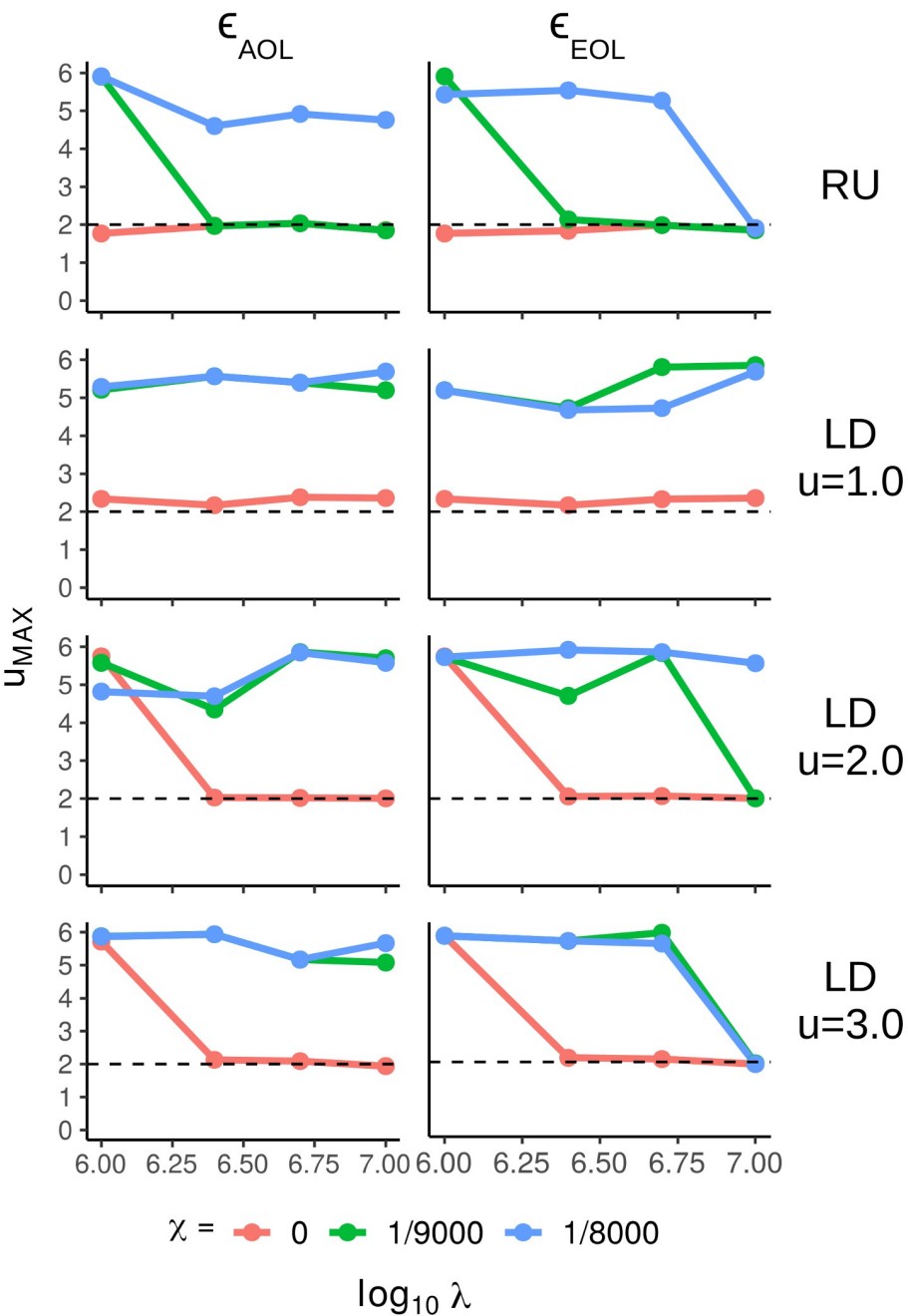

**Fig 7. Optimal foraging exponents.** The exponent which maximizes fitness, $u_{MAX}$, extracted from the sliding window of the positive-only first moments from the $5 \cdot 10^4$ after applying searching costs ($\chi = 0$ [pink], $\chi = 1/9000$ [green], $\chi = 1/8000$ [blue]), as a function of $\log_{10}$ lifespan $\lambda$. The left column is the average of energy over a lifespan, $\epsilon_{AOL}$, the right column is energy at the end of a lifespan, $\epsilon_{EOL}$, and the rows are indexed by the resource distribution type (RU = Random Uniform, LD = Lévy Dust where $u - 1$ = fractal dimension).

The results of the single-generation simulations demonstrated that most scenarios would result in selection for DOs with Lévy-like exponents, particularly with longer lifespans. However, they also demonstrated a potential interplay between the large fitness variance of Brownian-like DOs and the higher mean fitness of Lévy-like DOs when lifespans are short, or when

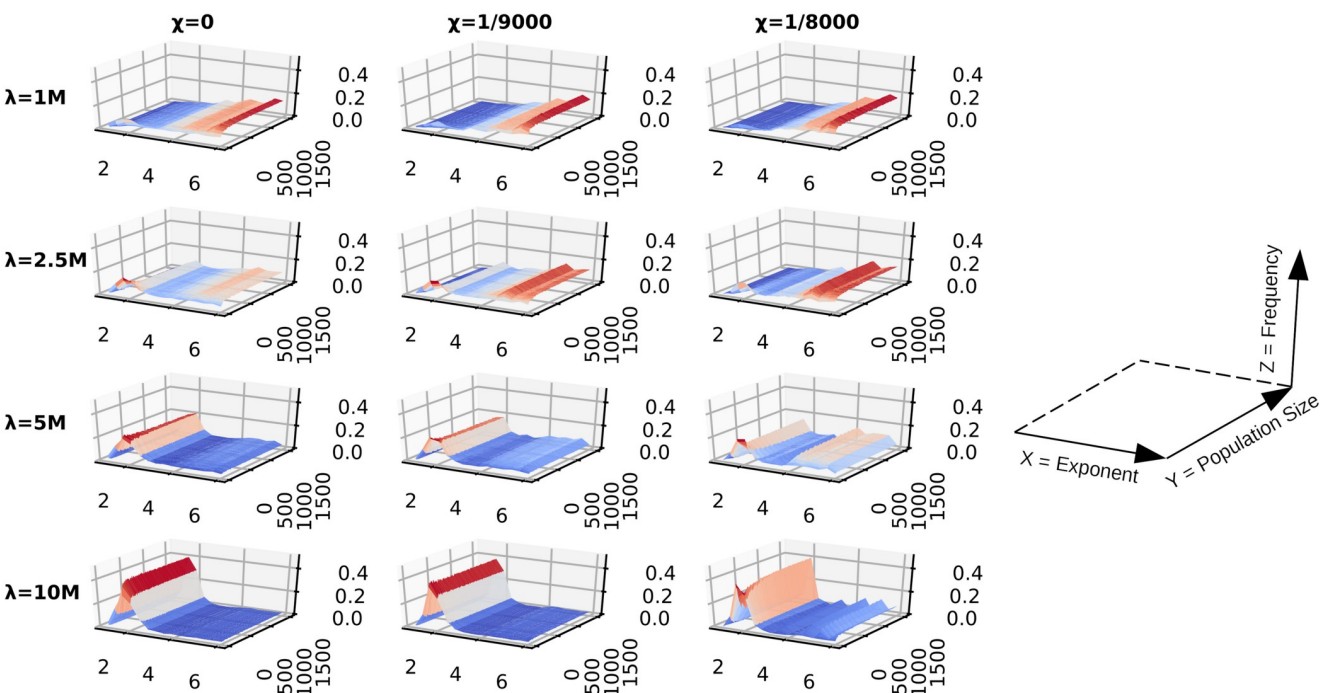

**Fig 8. DO fitness landscape with random uniform resources.** A matrix of surface plots of the top 1% performing DOs from sub-populations of the single-generation simulations which traversed random uniform environments. Fitness is determined by the average energy over a lifespan, $\epsilon_{AOL}$. The matrix rows are indexed by lifespan $\lambda$, and the columns by the searching cost $\chi$. A reference plot has been provided in order to interpret the axes of the individual surface plots; the x-axis is foraging exponent, the y-axis is population size, and the z-axis is frequency normalized to [0, 1].

searching in ballistcally distributed resources. Therefore, we primarily focused on comparing evolutionary simulations with no searching costs ($\chi = 0$) and high costs ($\chi = 1/8000$), small ($K = 10$) and large ($K = 1500$) population size, lifespans of $\lambda = 1M$, and across all resource distributions. As natural selection is predicted to result in Lévy-like behaviour, the $0^{th}$ generation of each run began with a population of DOs with a mean foraging exponent of either $u = 1.0$ or $u = 5.0$ with the predicted outcome that, after several generations, the mean foraging exponent would approach $u = 2.0$. The alternative, the extrinsic hypothesis, is that selection will result in foraging exponents that are positively correlated with the spatial exponent governing the distribution of resources.

## Evolutionary simulations

The evolutionary simulations with no cost resulted in overwhelming selection and evolution of Lévy-like DOs (Fig 11A), and demonstrated a negative correlation between evolved foraging exponent and Lévy dust dimension (Fig 12), but there is some nuance. The mean fitness of the surviving DOs increased, but the number which survived decreased, with increasing clumpiness of resources. However, the mean fitness of Brownian-like DOs exceeded Lévy-like DOs, with fewer surviving DOs for LD dimensions $u = 2.0$ & $3.0$. This provides evidence that Lévy-like behaviour acts to maximize the, long-term, or geometric-mean fitness, and may thus evolve as bet hedging [48, 71], a strategy known to be especially important in small populations [72]: although Brownian-like behaviour can result in the highest fitness individuals within a generation, geometric-mean fitness is reduced because survival is variable, whereas Lévy-like behaviour results in higher reliability of survival, thus maximizing geometric-mean fitness

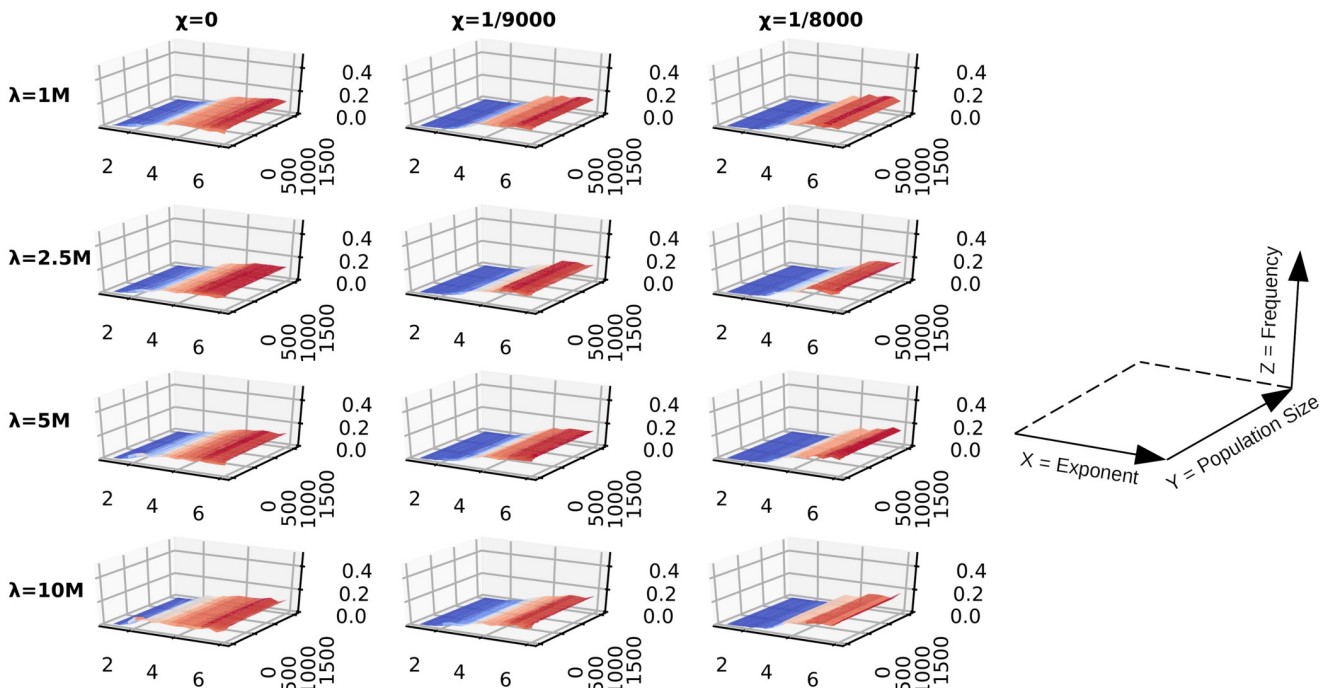

**Fig 9. DO fitness landscape with LD $u$ = 1.0 resources.** A matrix of surface plots of the top 1% performing DOs from sub-populations of the single-generation simulations which traversed Lévy dust environments with dimension $u$ = 1.0. Fitness is determined by the average energy over a lifespan, $\epsilon_{AOL}$. The matrix rows are indexed by lifespan $\lambda$, and the columns by the searching cost $\chi$. A reference plot has been provided in order to interpret the axes of the individual surface plots; the x-axis is foraging exponent, the y-axis is population size, and the z-axis is frequency normalized to [0, 1].

[73]. However, there was one exceptional set of simulations: small population size with LD dimension $u$ = 1.0 environments. Small populations are highly subject to drift, and while many runs resulted in Lévy-like behaviour (25/40 runs for $\sigma_{sv}$ = 0.25, 12/40 runs for $\sigma_{sv}$ = 0.50), a roughly equal number of runs evolved towards large positive, and even negative exponents, indicative of the flat fitness curve for LD dimension $u$ = 1.0 environments. The magnitude of the effect of drift, as well as selection, varied with the standing variation $\sigma_{sv}$: larger standing variation amplifies the effects of drift and speed in which evolution transpired. Including a high searching cost resulted in overwhelming selection for Brownian-like DOs for large population sizes (Fig 11B), but Lévy-like DOs for small population sizes, except for LD dimension $u$ = 1.0 environments. It should be noted that most small population simulations with cost resulted in extinctions, and selection in this case means a foraging exponent persisted for tens to hundreds of generations longer relative to another foraging exponent. The simulations with cost accentuated some of the fitness differences of the zero-cost simulations. Brownian-like DOs always had higher mean fitness, but Lévy-like DOs (especially foraging exponent $u$ = 2.0) still had the highest number of surviving DOs, again, with the exception of LD dimension $u$ = 1.0 environments. These results prompted us to dig deeper in two of the more polarizing results; the Brownian dominance in LD dimension $u$ = 1.0 environments and the larger number of offspring survival for Lévy-like DOs in LD dimension $u$ = 3.0 environments.

The two final sets of simulations with high cost were 1) evolutionary simulations with lifespans increased from $\lambda$ = 1M to $\lambda$ = 2.5M, and with large population size, and 2) competition simulations of equal-sized populations of fixed exponents, Brownian-like ($u$ = 5.0) and Lévy-like ($u$ = 2.0) DOs for large ($K$ = 3000) and small ($K$ = 20) population sizes. Longer lifespans resulted in the selection and evolution for Lévy-like exponents despite the high searching cost

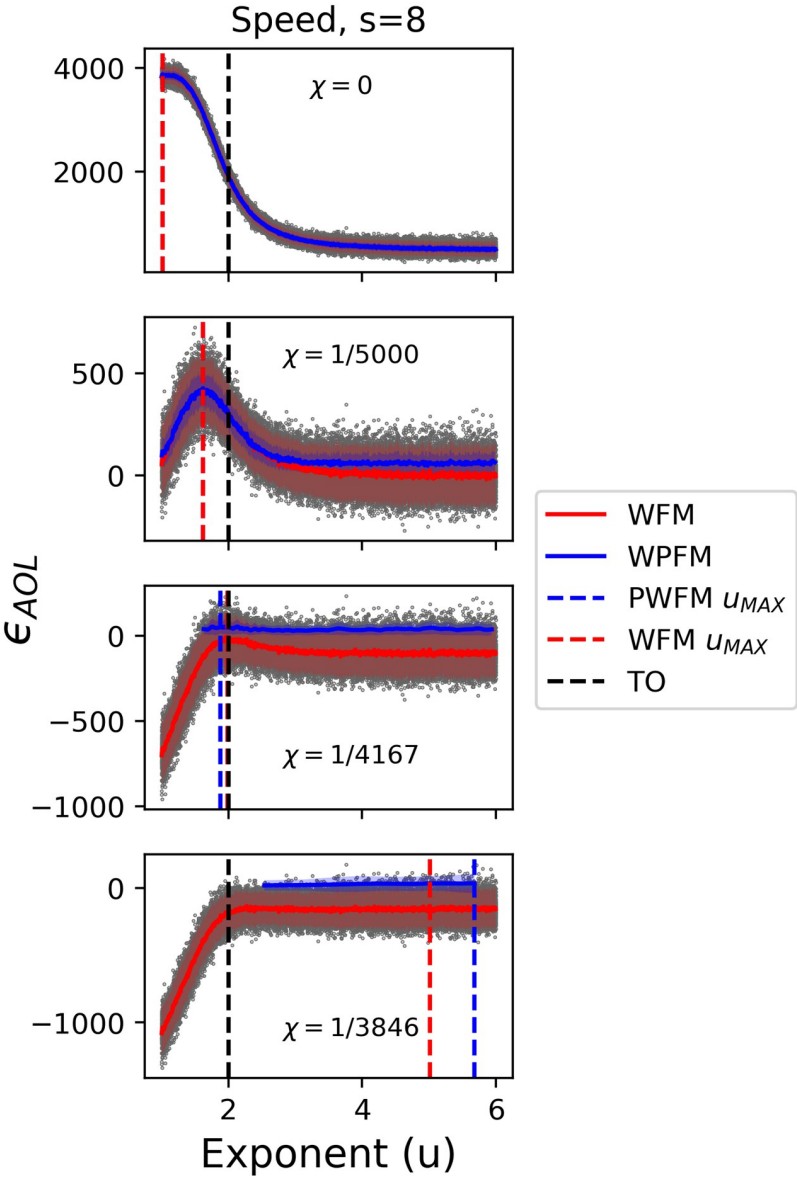

**Fig 10. Increasing the speed of DOs results in ballistic optimums.** The final average energy over a lifespan $\epsilon_{AOL}$, of 5 $\cdot\ 10^4$ DOs with foraging exponents $1 \leq u \leq 6$, lifespans $\lambda = 5M$, traveling with speed $s = 8$, in random uniform environments, and with an increasing searching cost $\chi$, by row. The fitness trends were captured with a sliding window of the first moment (FM) and positive-only first moment (PFM), with two standard deviations surrounding the trend. The exponent which maximizes fitness, $u_{MAX}$, is extracted from the sliding windows.

and the results of the single-generation simulations—indicating the importance of simulating evolutionary mechanisms and ecological contexts when making evolutionary predictions. Interestingly, the distribution of foraging exponents differed between the LD $u = 1.0$ and $3.0$ environments (Fig 13). Whereas $\sim 85\%$ of foraging exponents fell within the Lévy-like zone of $1 < u < 3$ for the final generation in LD dimension $u = 3.0$ environments, only $\sim 60\%$ did for LD dimension $u = 1.0$, with comparatively heavy skew into Brownian-like foraging exponents. This is indicative of the flatter fitness curves associated with increased resource homogeneity. The competition simulations consistently selected for Lévy-like DOs in LD dimension

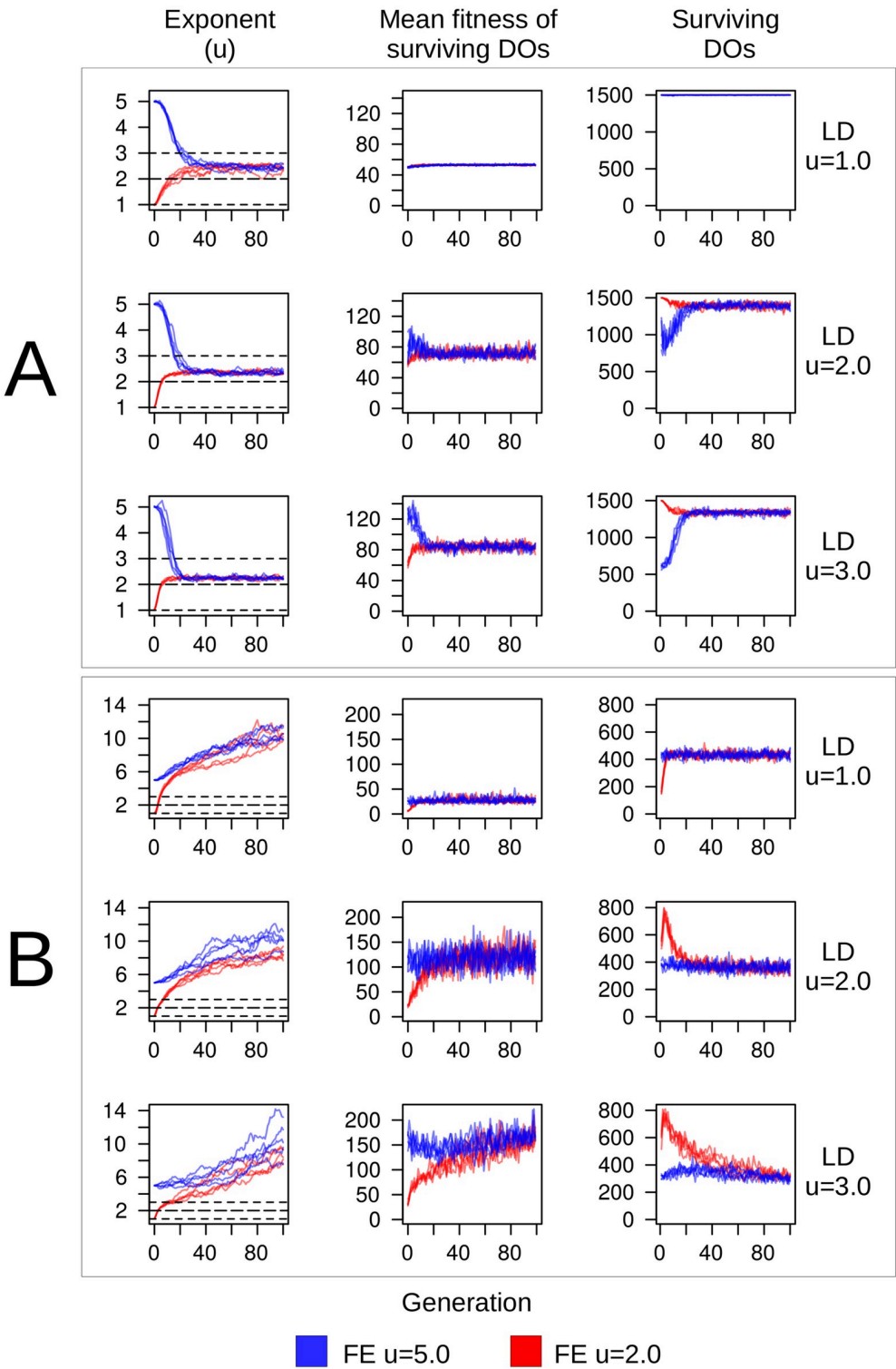

**Fig 11. Evolution of DO foraging exponents.** Evolutionary simulations of DOs with lifespans of $\lambda$ = 1M over 100 generations. (A) Searching costs $\chi$ = 0. (B) Searching costs $\chi$ = 1/8000. Each row comprises ten runs of an initial population of 1500 DOs, five runs starting with mean foraging exponent (denoted as FE) $u$ = 1.0 (red), the remaining five with $u$ = 5.0 (blue), both with $\sigma_{sv}$ = 0.5. Each row also represents a different environment (RU = Random Uniform, LD = Lévy Dust where $u-1$ = fractal dimension). Note: we exclude the random uniform results as they were simply an intermediate result between the LD $u$ = 1.0 and $u$ = 2.0 results.

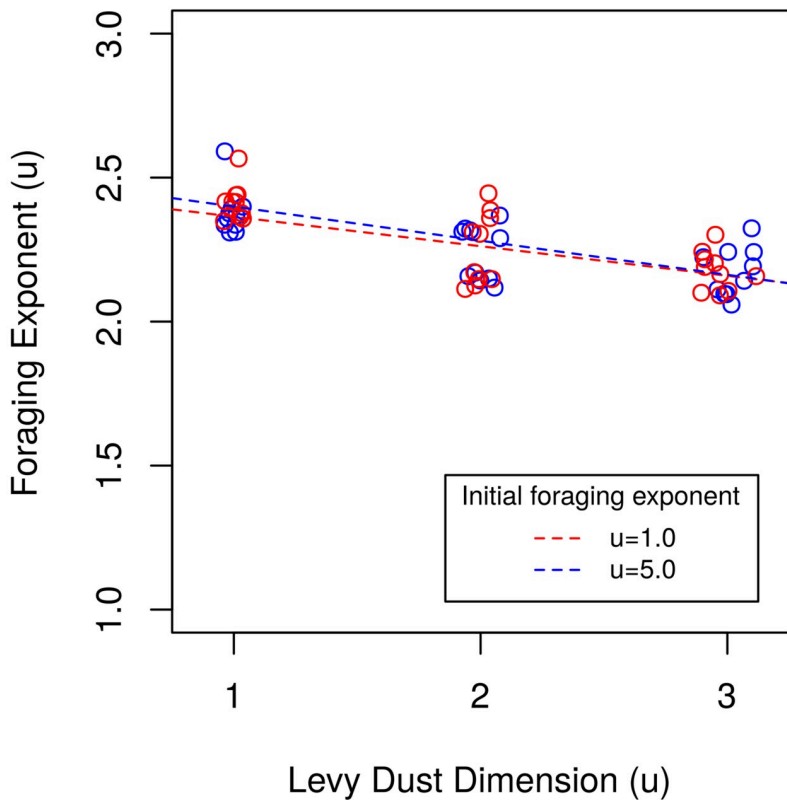

**Fig 12. Testing the extrinsic hypothesis.** Determining whether the evolved foraging exponents from simulations with $K = 1500, \chi = 0$, and $\sigma_{sv} = (0.25, 0.5)$ positively correlate with the Lévy dust dimension. Simulations of populations with an initial mean foraging exponent of $u = 1.0$ are marked with red circles, $u = 5.0$ with blue circles, and their trends are marked with a colour corresponding dashed line; u = 1.0: $r = -0.70$ ($P < 0.05$, $t_{df=28} = -5.2$), u = 5.0: $r = -0.74$ ($P < 0.05$, $t_{df=28} = -5.9$). Note: jitter was added to the x-axis to help differentiate points.

$u = 3.0$ (Fig 14, right column), contrary to the evolutionary simulations, and for Brownian-like DOs in LD dimension $u = 1.0$ (Fig 14, left column), in agreement with the evolutionary simulations. Performance when in competition is an important component of evolution, and thus provides an additional context necessary for making evolutionary predictions. In this case, Brownian foraging behaviour would be outcompeted by Lévy behaviour, even with large population size in environments with clumpy resources. But with short lifespans and a homogeneous distribution of resources, Brownian behaviour is advantageous.

Clearly, most scenarios result in the selection of Lévy-like exponents, but the outcome of selection is conditional on the length of a lifespan, whether the search is costly, the resource distribution, and the population size. The only results in opposition to the Lévy flight foraging hypothesis were contingent on a searching cost and shorter lifespans; thus, the realism of those parameters, and their predictions, should be addressed. We found that costs $\chi > 10^{-4}$ were sufficient to shift the outcomes of selection towards Brownian-like DOs, and our simulations happened to have environments with resource densities of $10^{-4}$ (i.e. $10^2$ resources entries within $10^6$ locations). In fact, a few test simulations determined that the effect of a cost remains the same after a 1:1 scaling with resource density, at least up to 10-fold density. A DO with a lifespan of $\lambda = 1M$, and speed $s = 1$, can theoretically visit every single location in our $1000 \times 1000$ matrices exactly once, in which case a cost of $\chi = 10^{-4}$ would result in a fitness of exactly zero.

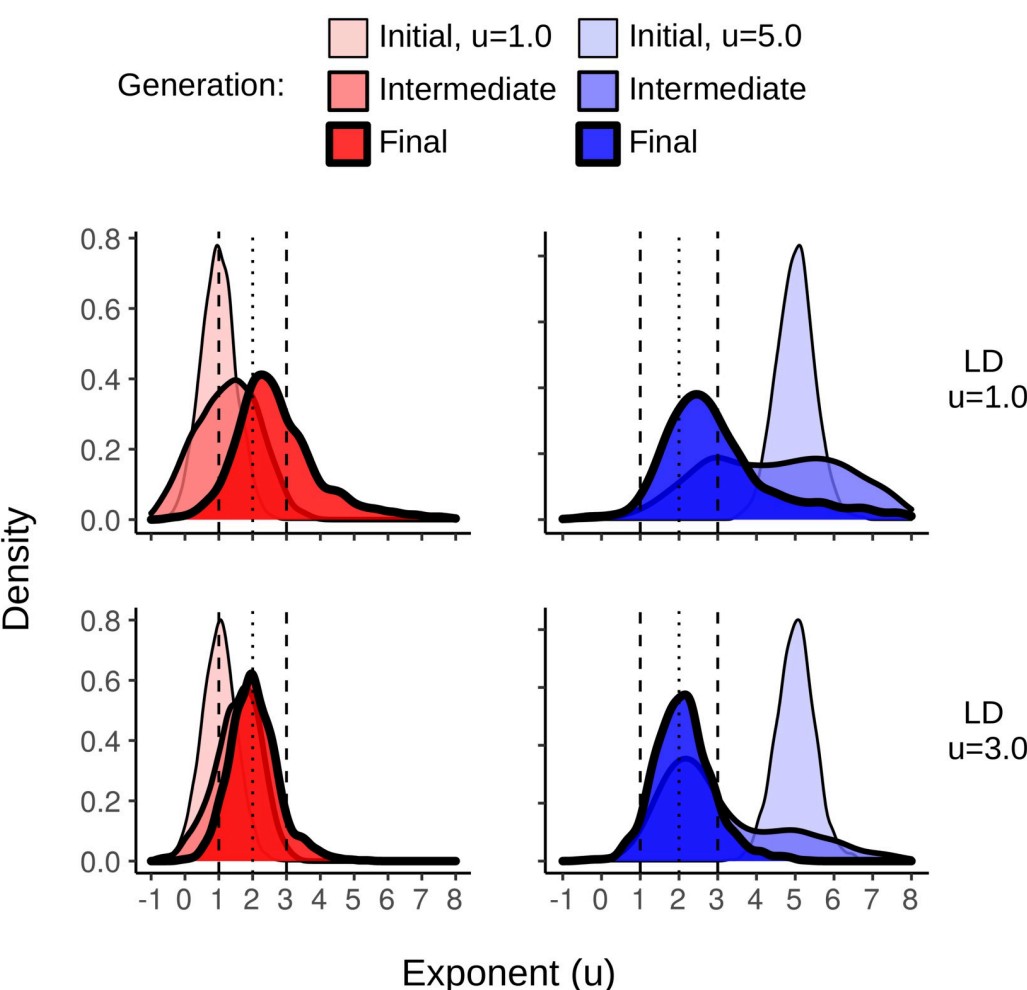

**Fig 13. Increasing the length of a lifespan selects for Lévy-like exponents.** The density curves of foraging exponents for the initial, intermediate, and final generations of evolutionary simulations with lifespans of $\lambda = 2.5M$ and a searching cost $\chi = 1/8000$. The first generations were composed of 1500 DOs with mean foraging exponents $u = 1.0$ or $u = 5.0$, and $\sigma_{sv} = 0.5$. The top row is the results from a LD dimension $u = 1.0$ resource distribution, the bottom LD dimension $u = 3.0$.

Therefore, Brownian foraging behaviour is predicted if the cost to search an environment is greater or equal to the resource density, and if the population size is sufficiently large to produce high-performing individuals—assuming the resources are re-visitable and the total distance searched is close to spanning the environment. Organisms with Brownian behaviour have a higher probability of re-visiting the same resource and their behaviour may be optimal for shorter lifespans, but with sufficient time they would diffuse in to empty space reducing the probability of re-visits. A Lévy-like exponent, however, would have a higher probability of leaving that empty space and eventually encountering resources; exactly why the fitness variance in Brownian-like DOs decreases with increasing lifespan. Similarly, this explains why Brownian DOs had higher mean fitness in clumpy environments, but a higher number of surviving offspring in uniform environments, when lifespan was short. These results are in agreement with those of Dannemann's Lotka-Volterra models, where the most resilient predators (least likely to face extinction) utilized a Lévy like movement strategy, with $u \simeq 2$ [74].

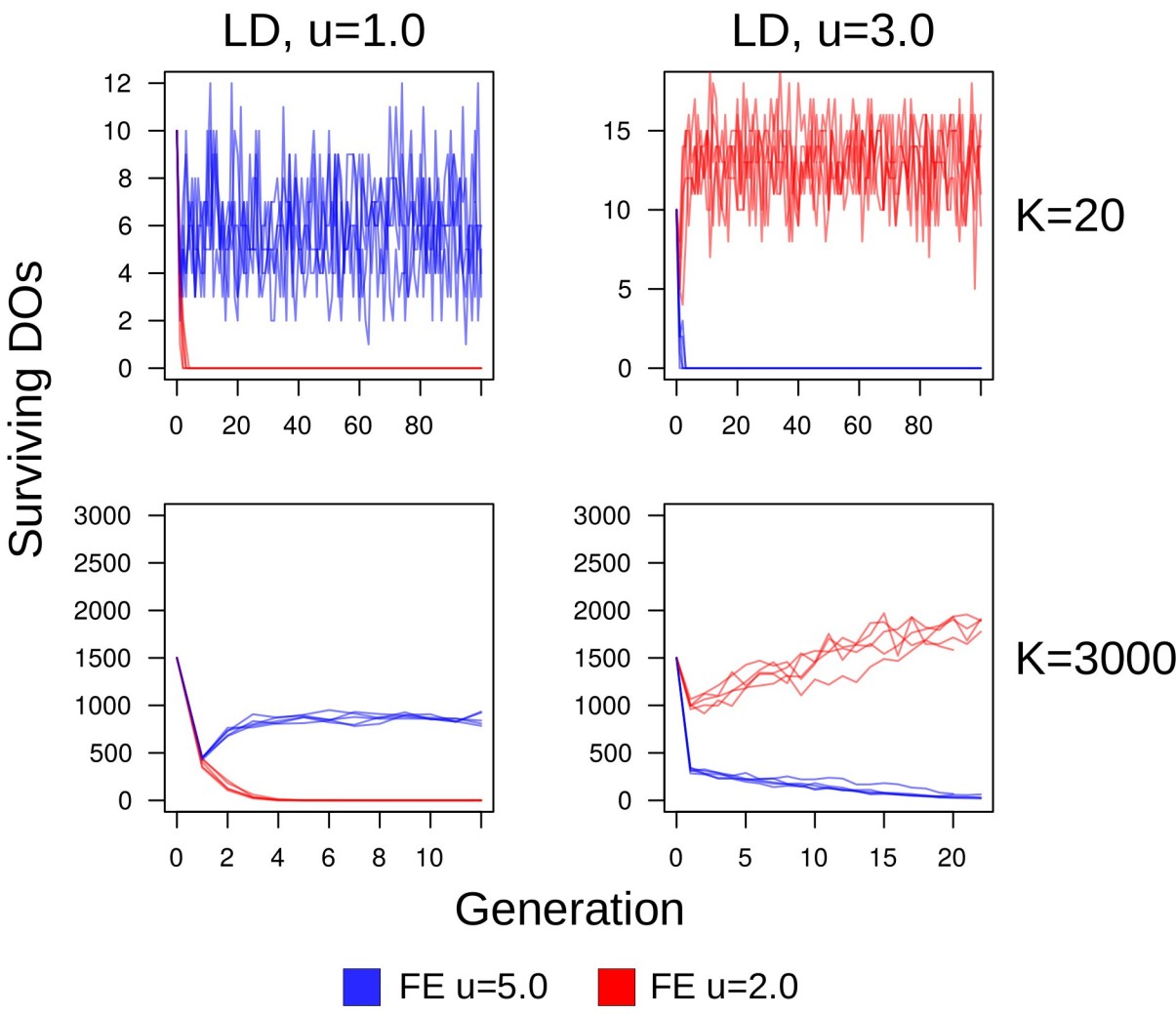

**Fig 14. Competition between fixed Lévy and Brownian foraging exponents.** Competition simulations of equal-sized populations with fixed foraging exponents (denoted as FE), lifespans λ = 1M, and searching cost χ = 1/8000; Lévy-like $u$ = 2.0 (red) versus Brownian-like $u$ = 5.0 (blue). The carrying capacity of the top row is $K$ = 20 DOs, the bottom row is $K$ = 3000 DOs. The left column is DOs traversing LD environments with dimension $u$ = 1.0, and the right column is fractal dimension $u$ = 3.0.

## Conclusion

These results provide evidence that the intrinsic, or adaptationist, hypothesis is a sufficient explanation for Levy-like behaviour: Lévy flight can be the result of selection for behavioural adaptations, rather than an emergent phenomenon due to the encounters within an environment's distribution of resources. Indeed, there are examples of emergent Lévy-like behaviour, such as Boyer's model of primate foraging, which corresponded with empirical evidence of spider monkey foraging patterns [75], or deterministic walks resulting in self-avoiding behaviour [76]. However, these models deviate from the assumptions of the Lévy flight foraging hypothesis, and are concerned with non-evolutionary explanations; we find no support for the extrinsic, or emergentist, hypothesis within the purview of our evolutionary models. The fitness of Lévy-like foraging is realized over longer timespans, be it over a lifespan or multiple generations, and with the evidence that the behaviour spans multiple taxa, Lévy flight foraging is perhaps a deep, evolutionarily conserved trait [48]. Memory and sensory adaptations most likely

trump a random walk in most scenarios. But, given that environmental variation is inevitable, Lévy-like behaviour is a searching strategy which offers a sufficient amount of success regardless of that variation, especially for cases where environmental cues, or memory, may be ineffective. We provided no evidence here on de novo distributions of move lengths resembling Lévy-like behaviour, or selection due to variable environments (i.e. temporally varying the density and/or distribution of resources). Such studies, empirical or theoretical, will be be instrumental in providing further evidence for the Lévy flight foraging hypothesis, and in determining whether Lévy-like foraging behaviour is a form of bet-hedging. We hope the results provided here will motivate further exploration into determining the ultimate explanation for Lévy flight foraging.

## Supporting information

**S1 Fig. Fitness curve of DOs traversing random uniform environments.**
(TIF)

**S2 Fig. Evolution of DO foraging exponents, small population size.**
(TIF)

**S1 Appendix. Detailed proof of Box 1.**
(PDF)

**S1 Code. Code for single-generation and evolutionary simulations.**
(ZIP)

**S1 Data. Metadata and results of the single-generation and evolutionary simulations.**
(ZIP)

## Acknowledgments

Thank you to R. Taylor for providing computational resources and IT advice, and to S. Raghu, L. Wen, H. Hitsman, and A. Cheslock for support and guidance.

## Author Contributions

**Conceptualization:** Winston Campeau, Andrew M. Simons, Brett Stevens.

**Data curation:** Winston Campeau.

**Formal analysis:** Winston Campeau, Brett Stevens.

**Funding acquisition:** Andrew M. Simons, Brett Stevens.

**Investigation:** Winston Campeau.

**Methodology:** Winston Campeau, Andrew M. Simons, Brett Stevens.

**Project administration:** Winston Campeau.

**Resources:** Winston Campeau, Brett Stevens.

**Software:** Winston Campeau.

**Supervision:** Andrew M. Simons, Brett Stevens.

**Validation:** Winston Campeau.

**Visualization:** Winston Campeau.

**Writing – original draft:** Winston Campeau.

**Writing – review & editing:** Winston Campeau, Andrew M. Simons, Brett Stevens.

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
