## [Decision Letter · Decision Letter 0]

16 Dec 2021

Dear Mr. Campeau,

Thank you very much for submitting your manuscript "The evolutionary maintenance of Lévy flight foraging: a numerical simulation." for consideration at PLOS Computational Biology. As with all papers reviewed by the journal, your manuscript was reviewed by members of the editorial board and by several independent reviewers. The reviewers appreciated the attention to an important topic. Based on the reviews, we are likely to accept this manuscript for publication, providing that you modify the manuscript according to the review recommendations.

Sincerely,

Marcos Gomes Eleuterio da Luz, Ph.D.

Guest Editor

PLOS Computational Biology

James O'Dwyer

Deputy Editor

PLOS Computational Biology

[LINK]

Reviewer's Responses to Questions

**Comments to the Authors:**

Reviewer #1: It is widely accepted that Levy flights and walks can optimize random

searches. The Levy flight foraging hypothesis holds that therefore

natural selection must have led to adaptations for Levy walks, etc.

This is an old problem that has already been studied extensively.

However, this is not the problem posed by the authors here. Instead,

they ask a different but related question, viz., about the

"maintenance of Lévy flight foraging through evolutionary processes."

The main result is that, based on numerical simulations, the authors

report evidence in favor of the adaptationist viewpoint, rather then

the emergentist explanation for observed Levy walks. The authors

report zero evidence for the latter. By studying multipled

generations, it is shown that Levy-type behavior is "is perhaps a

deep, evolutionarily conserved trait." In addition to the main text,

there are supplementary materials (including figs S1 and S2 and an

appendix, as well as zipped archives).

The results are quite impressive and I recommend publication as is.

This is a very nice paper indeed.

There is only one point that I did not understand enough: the use of

geometric rather than arithmetic means. We all know that there are

different kinds of means, including harmonic means. But after page 5

there is no more mention of geometric means until page 22. If the

authors can explain this point better it would be quite useful to the

reader.

Minor points:

** Is the terminology of intrinsic vs. extrinsic hypotheses new? This

should be clearly stated. I have only heard of these previously

referred to as the adaptationist vs. the emergentist viewpoints,

and so forth.

** The page number for ref. 71 is 182--211. I did not know this paper

and found it very interesting to read.

** As mentioned above was not able fully to understand the difference

between geometric and arithmetic mean fitness in this context.

Perhaps the authors can further explain this a little bit better.

Reviewer #2: The evolutionary maintenance of Lévy flight foraging: a numerical simulation.

PCOMPBIOL-D-21-01701

This is without doubt an interesting modelling exercise where the authors give "evolutionary" support to the Lévy Flight Foraging Hypothesis based on random searching when resources are sparse and no memory is involved.  The main results are that Lévy Flights are often the selected outcome  of a sort of genetic algorithm. However Brownian motion overcomes LF, sometimes. This fact is well addressed and necessary explanation is given when this is the case. The paper is well written.

I have some minor comments that must be addressed before this paper can be accepted. 

1. It is said in the abstract that  "Lévy flight is a type of random walk that models the behaviour of many phenomena across a multiplicity of academic disciplines''. I disagree with this sentence since LF are a natural phenomenon and not models. I suggest instead "Lévy flight is a type of random walk present in the behaviour of many natural phenomena across a multiplicity of academic disciplines."

2. Even though the possibility of the emergent nature of Lévy flight is mentioned, its treatment is somewhat sloppy. I suggest in order to give more punch to the paper and to show that the authors handle well the literature, that the following papers on deterministic walks be read and discussed either at the introduction or the conclusions:[Ref1] Lima, Gilson F., Alexandre S. Martinez, and Osame Kinouchi. "Deterministic walks in random media." Physical Review Letters 87.1 (2001): 010603.[Ref2] Boyer, Denis, et al. "Scale-free foraging by primates emerges from their interaction with a complex environment." Proceedings of the Royal Society B: Biological Sciences 273.1595 (2006): 1743-1750.

3. Line 425: "Organisms with Brownian behaviour have a higher probability of re-visiting the same resource and their behaviour may be optimal for shorter lifespans, 

but with sufficient time they would diffuse into empty space reducing the probability of 

re-visits.  A Lévy-like exponent, however, would have a higher probability of leaving that 

empty space and eventually encountering resources." There is a paper demonstrating this fact but that has been ignored:[Ref3] Dannemann, Teodoro, Denis Boyer, and Octavio Miramontes. "Lévy flight movements prevent extinctions and maximize population abundances in fragile Lotka–Volterra systems." Proceedings of the National Academy of Sciences 115.15 (2018): 3794-3799.

4. First sentence of the conclusion is misleading and there is indeed lack of general evidence through the paper to claim this as true: "These results provide evidence that the intrinsic (please correct the typo) hypothesis is a sufficient explanation for Levy-like behaviour: Lévy flight is the result of selection for behavioural adaptations, rather than an emergent phenomenon due to the encounters within an environment’s distribution of resources." [Ref2] gives enough theoretical evidence that the distribution of resources modulates the searching strategy resulting in emergent movement patterns. Perhaps the authors would like to explain under which particular and specific circumstances their claim is true and tone it down consequently.

**Have the authors made all data and (if applicable) computational code underlying the findings in their manuscript fully available?**

Reviewer #1: None

Reviewer #2: Yes

PLOS authors have the option to publish the peer review history of their article (what does this mean?). If published, this will include your full peer review and any attached files.

Reviewer #1: No

Reviewer #2: No

Figure Files:

Data Requirements:

Reproducibility:

References:

---

## [Editor Report · Decision Letter 1]

28 Dec 2021

Dear Mr. Campeau,

We are pleased to inform you that your manuscript 'The evolutionary maintenance of Lévy flight foraging' has been provisionally accepted for publication in PLOS Computational Biology.

Best regards,

Marcos Gomes Eleuterio da Luz, Ph.D.

Guest Editor

PLOS Computational Biology

James O'Dwyer

Deputy Editor

PLOS Computational Biology

---

## [Editor Report · Acceptance letter]

13 Jan 2022

PCOMPBIOL-D-21-01701R1 

The evolutionary maintenance of Lévy flight foraging

Dear Dr Campeau,

I am pleased to inform you that your manuscript has been formally accepted for publication in PLOS Computational Biology. Your manuscript is now with our production department and you will be notified of the publication date in due course.

With kind regards,

Olena Szabo
